# Dairy Buffalo Life Cycle Assessment (LCA) Affected by a Management Choice: The Production of Wheat Crop

Elio Romano [1] , Pasquale De Palo [2] , Flavio Tidona [3] , Aristide Maggiolino [2] and Andrea Bragaglio [2,*]

[1] Council for Agricultural Research and Economics, Research Centre for Engineering and Agro-Food Processing, CREA-IT, 24047 Treviglio, Italy; elio.romano@crea.gov.it

[2] Department of Medicine Veterinary, University of Bari "Aldo Moro", 70010 Valenzano, Italy; pasquale.depalo@uniba.it (P.D.P.); aristide.maggiolino@uniba.it (A.M.)

[3] Council for Agricultural Research and Economics, Research Centre Animal Production and Aquaculture, CREA-ZA, 26900 Lodi, Italy; flavio.tidona@crea.gov.it

* Correspondence: andrea.bragaglio@uniba.it

**Abstract:** Life cycle assessment (LCA) was performed in dairy buffalo farms representative of Southern Italian farming systems, similar due to several characteristics, with the exception of wheat production. This work evaluated the impacts derived from this management choice, comparing farms with wheat crop (WWC) or not (NWC). In agreement with the literature, economic allocation was chosen as a useful strategy to attribute equivalents to by-products, i.e., culled animals; the same criterion was also adopted to assign pollutants to wheat grain, limited to WWC farms. Environmental impacts in terms of Global Warming Potential (GWP, kg $CO_2$ eq), Acidification Potential (AC, g $SO_2$ eq), Eutrophication Potential (EU, g $PO_4^{3-}$ eq), Agricultural Land Occupation (ALO, $m^2y$) and Water Depletion (WD, $m^3$) were estimated. The production of wheat crop significantly affected ($p < 0.05$) the Agricultural Land Occupation (ALO) category as WWC farms need adequate land. WWC farms could allow a significant reduction in eutrophication (EU) compared to NWC farms ($p < 0.05$).

**Keywords:** life cycle assessment; dairy buffalo; forages; wheat crop; allocation





## 1. Introduction

Water buffalo (*Bubalus bubalis*) is a species that provides multiple products worldwide such as draught power, meat, milk, skin and manure. In Italy, buffalo farming has been conducted for centuries in extensive conditions based on marshland environments. Nowadays, most of the Mediterranean Italian Buffalo breed buffaloes are reared under intensive conditions to produce milk, almost entirely processed in mozzarella cheese. The production of buffalo mozzarella cheese is steadily increasing, +26.1% from 2013 to 2017, driven by the constant export increase, especially in Europe, the United States of America and Great Britain, and by the growing international interest in this product, as witnessed by the recent introduction of dairy buffalo in Germany and Great Britain [1–3].

Most of these animals are reared in the Protected Designation of Origin (PDO) area comprising the Campania, Lazio, Apulia and Molise regions, where recently buffalo husbandry has moved to more intensive farming conditions with a feeding system based on three different rations corresponding to the three main buffalo productive stages, i.e., lactating cows, dry cows and growing heifers [4]. Berlese et al. [1] also highlighted that in these conditions the animals have no access to pasture and water for wallowing.

Additionally, those areas are historically vocated to produce wheat, mainly to manufacture pasta, which is exported worldwide.

In Northern Italy, mozzarella cheese is sold directly to consumers through a short supply chain. In this context, dairy plants started transforming buffalo milk not exclusively

in mozzarella cheese but also in other products, such as ripened cheeses and ricotta, which are appreciated by consumers [1].

Although there is growing attention on the environmental impact of livestock farming, to our knowledge, few studies focused on milk buffalo are available. In particular, few LCA studies have been developed on the allocation of a by-product such as wheat grain (*Triticum durum Desf.*).

As indicated by several authors [5–7], dairy systems often produce crop commodities (cereals), thus in cases of multifunctional processes the environmental impact should be shared among products. The aim of this study was to compare the environmental impact of buffalo milk, provided with two different management systems, integrated with economic allocation.

## 2. Materials and Methods

### 2.1. The Farms

All the farms of this study, located in the Apulia and Basilicata regions, are specialized in buffalo dairy farming, with animals kept in confinement. They adopt sexed semen mainly for the heifers to keep high-value animals, and their profiles are described in Table 1. The primary data were obtained from six farms distributed in two groups: No Wheat Crop (NWC) and With Wheat Crop (WWC). Despite the fact that the feeding of both groups is balanced with a 2:1 ratio = corn silage:wheat/straw, only the WWC group does not purchase supplementary forages, such as hay and straw. The WWC farms had different arable lands but a similar incidence of wheat income. The NWC1, WWC2 and WWC3 farms are located in the Protected Designation of Origin (PDO) area of Foggia Province (Apulia Region), whereas Basilicata (Potenza and Matera Provinces) is excluded from the PDO. According to widespread knowledge, Mediterranean Italian Buffalo breed animals show a noticeable heterogeneity in size, and this characteristic would affect management, feeding and productions, also conditioning dry matter intake of feed and milk yields.

**Table 1.** Profile of the farms and of the two groups. Urea 46%N and ammonium nitrate 27%N.

| | No Wheat Crop (NWC) | | | With Wheat Crop (WWC) | | |
| --- | --- | --- | --- | --- | --- | --- |
| | NWC1 | NWC2 | NWC3 | WWC1 | WWC2 | WWC3 |
| Geographical place (Province) | Foggia | Potenza | Potenza | Matera | Foggia | Foggia |
| Total crop area, Ha | 20 | 65 | 80 | 65 | 225 | 270 |
| Hay, Ha | 20 | 50 | 40 | 40 | 50 | 140 |
| Barley, Ha | - | - | 10 | - | - | - |
| Maize silage, Ha | - | 15 | 30 | - | 15 | 15 |
| Maize grain, Ha | - | - | - | - | 10 | 20 |
| Wheat, Ha | - | - | - | 25 | 150 | 95 |
| Herd, heads n. | 197 | 303 | 446 | 203 | 479 | 613 |
| Lactating cows, n. | 60 | 120 | 150 | 52 | 160 | 185 |
| Dry cows, n. | 25 | 50 | 80 | 60 | 180 | 185 |
| Heifers, n. | 90 | 80 | 185 | 50 | 100 | 175 |
| Young < 365 days, n. | 20 | 45 | 25 | 35 | 30 | 60 |
| Bulls, n. | 2 | 8 | 6 | 6 | 9 | 8 |
| Urea, t y$^{-1}$ | - | 10.0 | 20.0 | 37.5 | 45.0 | 59.5 |
| Ammonium nitrate, t y$^{-1}$ | - | - | 7.0 | - | - | - |
| Phosphate, t y$^{-1}$ | - | - | - | - | 0.8 | 1.6 |
| Potassium chloride, t y$^{-1}$ | - | - | 5.2 | - | - | - |
| Concrete area (shed, services), m$^2$ | 1500 | 4000 | 3500 | 1000 | 6000 | 11,000 |
| Milking parlor size, m$^2$ | 200 | 200 | 400 | 150 | 300 | 300 |
| Milk tank, liters | 1400 | 6000 | 5000 | 2500 | 4000 | 6000 |
| Diesel, liters y$^{-1}$ | 18,800 | 23,500 | 76,500 | 21,200 | 64,700 | 70,500 |
| Electricity, kWh y$^{-1}$ | 45,700 | 65,000 | 97,000 | 50,000 | 76,000 | 87,600 |

### 2.2. Life Cycle Assessment (LCA) Methodology

The farms were analyzed with an LCA approach, based on ISO 14040 and ISO 14044 methodology [8,9]. The principles and framework for LCA include four distinct phases: (1) definition of the goal and scope (including functional unit and limits of the system); (2) life cycle inventory (LCI) analysis (including input and output data collection for all processes); (3) life cycle impact assessment (LCIA); (4) life cycle interpretation.

#### 2.2.1. Goal and Scope Definition

In this study, the environmental impacts of milk obtained from two different farming managements in terms of Global Warming Potential (GWP, kg $CO_2$ eq), Acidification Potential (AC, g $SO_2$ eq), Eutrophication Potential (EU, g $PO_4^{3-}$eq), Agricultural Land Occupation (ALO, $m^2$y) and Water Depletion (WD, $m^3$) were assessed. The functional unit was 1 kg of normalized buffalo milk, with a reference milk fat and protein (fat and protein corrected milk, FPCM) content of 8.3 and 4.73%, respectively. Raw milk was transformed into FPCM with the following equation [10,11]:

$$\text{FPCM (kg/yr)} = (\{[(g \text{ of fat/L} - 83) + (g \text{ of protein/L} - 47.3)] \times 0.00687\} + 1) \times \text{milk production (kg/yr)} \quad (1)$$

#### 2.2.2. System Boundary Definition

The system boundaries considered in the study were comprised "from cradle to farm gate" (Figures 1 and 2). All the on-farm operations (e.g., animal feeding and care, milking procedures) and sources pertaining to forage production (e.g., arable land, agrochemicals, water) were considered. The consumption of energy and the emissions from activities conducted off-farm were retrieved from databases provided by SimaPro 8.03 [12]. The transport of off-farm feeds, fossil fuels and bedding materials and their emissions were also included in the assessment.

#### 2.2.3. Allocation Criterion

Dairy systems are typically multifunctional processes that coproduce meat from culled animals and surplus calves, for example, in addition to milk. In particular, the farms involved in this study are located in Southern Italian regions (Apulia and Basilicata), where the cultivation of wheat grain (*Triticum durum Desf.*) is largely widespread. This research focused on the presence of this peculiar coproduct, considered a relevant managing factor, to obtain two advantages for the three wheat-based farms:

1. A significant income in addition to milk.
2. The wheat also provides straw, adopted as fodder (mainly for dry cows) and litter. The With Wheat Crop (WWC) farms benefit from full forage self-sufficiency.

For these reasons, the economic allocation criterion was applied to both the by-product wheat grain and culled cows, in agreement with Pirlo et al. [11].

Young females are kept as replacers in the amount described by the owners, whereas male calves are sold at the 15th day of life; thus, the inputs and outputs provided by these animals were not considered.

#### 2.2.4. Inventory Analysis and Input Data

The inventory data on livestock production, crop cultivations, straw for litter, inputs of purchased feed, electricity, diesel consumption and farm extension (divided in crop Ha year$^{-1}$) were processed as primary data. The consumption and emission factor of natural gas adopted in the farms (water heating) was excluded due to the low impact and because it was not possible to have precise information about it. The feed consumption was retrieved by interviews with the farmers, the kind of water (i.e., well, tape) used in the farms was also obtained with interviews, whereas the consumption was assessed. In addition, a brief description about milk performances is reported in Table 2.

The inventory analysis involved the buildings, as sheds and concrete paved areas, whereas the structures and the bunkers (for corn silage) to stock the forages were excluded.

A productive life of 50 years was assumed for these constructions, as suggested by the Ecoinvent 3 allocation database. The on-farm feed was assumed to be transported for 1 km with a tractor (Transport, tractor and trailer, agricultural (GLO), market for, Alloc, Def, S) whereas the bought feed, carried by truck (Transport, freight lorry > 32 metric ton, EURO 5, RER), was computed based on the distance from the farm, as suggested by Bragaglio et al. [13]. The sunflower meal feed, in most cases, is also imported [14]; thus, a 2700 km travel distance by truck from Ukraine was assessed. According to other authors [15], a 10,000 km journey by ship (Transport, freight, sea, transoceanic ship (GLO), market for, Alloc, Def, S) for soybean and cotton seeds from South America and a 12,000 km journey for palm oil from Malaysia were assumed.

Finally, the journey from the Italian harbor to farm gate was added to this computation. For fossil fuel provision, we also considered transport by truck: transport freight lorry of 3.5–7.5 metric ton, EURO 5, RER.

The water consumption was distinguished in specific categories, provided by the software Simapro 8.03: the water needed for transoceanic crops, the Italian well water mainly adopted for maize and the water adopted for the industrial processes (for example, wheat flour shorts). A description of the formulation and composition of the diets administered by the six farms is shown in Tables 3 and 4, respectively, for cows (lactating and dry, LC and DC), heifers (HF) and young animals (91–365 days old, YA). The diets of the calves until the 90th day, based on milk replacers, weaning meal and a small amount of hay as sources of fiber, are very similar and were not reported in the table. Their amounts were loaded into the software and processed.

**Table 2.** Dairy performances and milk components of No Wheat Corn (NWC) and With Wheat Corn (WWC) farms.

| Item | Unit | NWC1 | NWC2 | NWC3 | WWC1 | WWC2 | WWC3 |
|---|---|---|---|---|---|---|---|
| FPCM yield | kg/year | 254,405 | 370,400 | 481,800 | 146,000 | 438,000 | 474,850 |
| Average FPCM per lactation * | kg/head | 3137 | 3087 | 2374 | 2076 | 2025 | 1897 |
| Fat mean ± SD | % | 7.38 ± 0.31 | 7.46 ± 0.45 | 7.83 ± 0.40 | 7.54 ± 0.41 | 8.30 ± 0.41 | 8.57 ± 0.53 |
| Protein mean ± SD | % | 4.63 ± 0.12 | 4.25 ± 0.16 | 4.48 ± 0.07 | 4.42 ± 0.21 | 4.65 ± 0.09 | 4.68 ± 0.14 |
| Bacterial count mean ± SD | cfu/mL | 76,670 ± 50,796 | 316,224 ± 25,215 | 131,667 ± 113,900 | 91,167 ± 37,649 | 73,077 ± 67,661 | 97,947 ± 47,339 |

* Assuming 270 and 95 days for lactation and dry period, respectively [4].

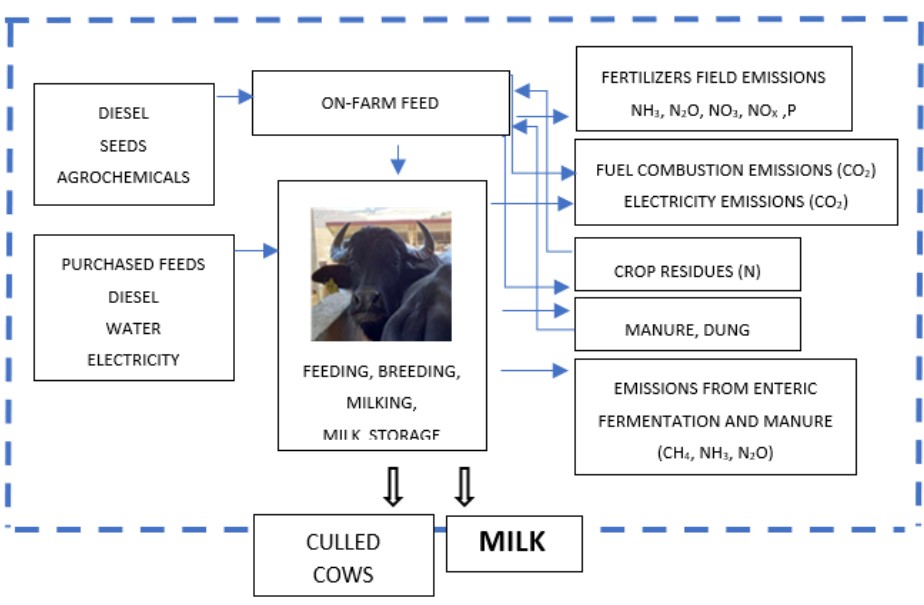

**Figure 1.** System boundaries of No Wheat Crop (NWC) system.

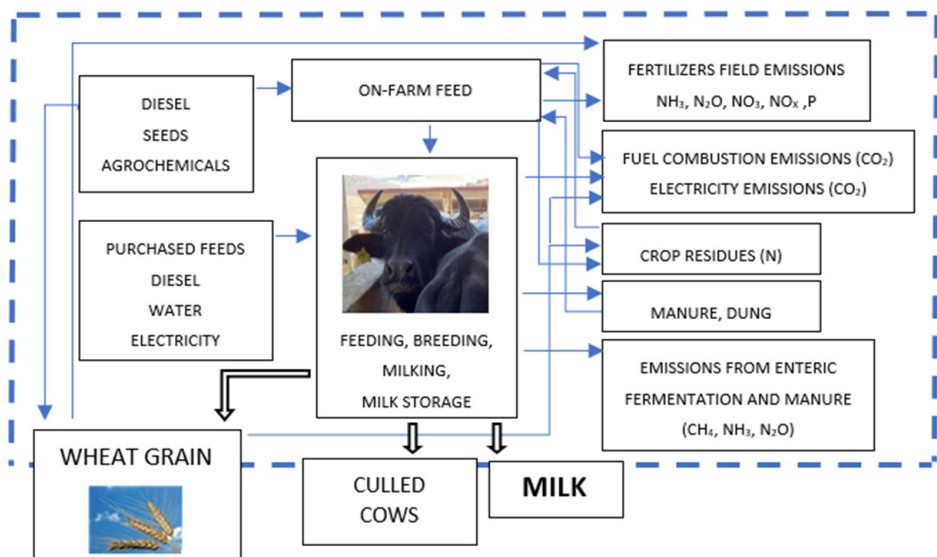

**Figure 2.** System boundaries of With Wheat Crop (WWC) system.

**Table 3.** No Wheat Crop (NWC) diets. Animal categories: lactating cows (LC), dry cows (DC), heifers (HF), young animals (YA); # Vicia faba minor; * kg/head/day.

| Category | NWC1 | | | | NWC2 | | | | NWC3 | | | |
|---|---|---|---|---|---|---|---|---|---|---|---|---|
| | LC | DC | HF | YA | LC | DC | HF | YA | LC | DC | HF | YA |
| **Forage kg/head/day** | | | | | | | | | | | | |
| Meadow hay | 8.0 | 5.5 | 3.5 | 2.0 | 10.0 | 7.0 | 5.0 | 3.0 | 3.2 | - | 3.2 | 1.5 |
| Alfalfa hay | 2.0 | - | 1.8 | - | - | - | - | - | - | - | - | - |
| Straw | - | 5.5 | - | - | - | 4.0 | 2.0 | 1.0 | - | 7.0 | 2.0 | - |
| Maize silage | - | - | - | - | 8.0 | 5.0 | 3.0 | 2.0 | 19.0 | 6.0 | 13.0 | 2.0 |
| **Raw concentrate kg/head/day** | | | | | | | | | | | | |
| Maize flour/grain | 4.0 | - | 1.3 | - | - | - | - | - | 4.0 | - | - | 0.5 |
| Barley | 1.5 | - | 0.4 | 1.0 | - | - | - | - | 2.0 | 2.0 | - | 1.0 |
| Soybean meal | 1.1 | - | - | - | - | - | - | - | - | - | - | - |
| Wheat flour shorts | - | 2.0 | - | - | - | - | - | - | - | - | - | - |
| **Market concentrate g/head/day** | | | | | | | | | | | | |
| Soybean seeds (roasted) | 310 | - | - | - | - | - | - | - | - | - | - | - |
| Soybean seeds (dehulled/flaked) | 240 | - | 480 | 400 | 2600 | - | 800 | 400 | 1650 | 260 | 200 | 150 |
| Sunflower meal | 480 | - | 440 | 360 | 2000 | - | 600 | 260 | 450 | 380 | 300 | 220 |
| Cotton seeds | 540 | - | - | - | 1600 | - | 500 | 220 | 450 | - | - | - |
| Maize flour | 450 | - | - | - | 300 | - | 100 | 80 | - | 130 | 100 | - |
| Fava bean # | 480 | - | - | - | - | - | - | - | - | - | - | - |
| Wheat flour shorts | 280 | - | 480 | 400 | - | - | - | - | - | 700 | 560 | 80 |
| Beet pulp | 280 | - | - | - | - | - | - | - | - | - | - | 350 |
| Linseeds | 240 | - | - | - | - | - | - | - | - | - | - | - |
| Wheat flour | 220 | - | - | - | - | - | - | - | - | - | - | - |
| Bran | - | - | 440 | 360 | - | - | - | - | 800 | 680 | 540 | 400 |
| Maize germ meal | - | - | 220 | 180 | - | - | - | - | - | 130 | 100 | - |
| Maize distillers | - | - | 60 | 50 | - | - | - | - | 800 | 130 | 100 | 80 |
| Palm oil | 60 | - | - | - | - | - | - | - | - | - | - | - |
| Molasses | 50 | - | 60 | 50 | - | - | - | - | - | 130 | 100 | 80 |
| **Chemical composition (%)** | | | | | | | | | | | | |
| Dry matter (DM) * | 17.8 | 11.7 | 8.2 | 4.4 | 18.1 | 11.4 | 8.3 | 5.0 | 18.2 | 12.1 | 10.5 | 4.6 |
| Crude protein (%DM) | 12.45 | 9.65 | 15.50 | 15.00 | 14.00 | 9.70 | 12.35 | 14.20 | 14.25 | 9.05 | 11.80 | 14.50 |
| Ether extract (%DM) | 5.80 | 3.50 | 3.45 | 3.70 | 3.15 | 2.75 | 2.90 | 3.40 | 4.40 | 3.25 | 3.15 | 4.15 |
| Crude fiber (%DM) | 22.70 | 36.60 | 28.00 | 18.70 | 27.60 | 35.90 | 26.65 | 32.20 | 21.40 | 29.85 | 25.20 | 18.95 |
| Ash (%DM) | 7.60 | 7.00 | 8.35 | 8.35 | 7.25 | 3.90 | 5.45 | 4.10 | 7.20 | 7.05 | 7.45 | 6.75 |

**Table 4.** With Wheat Crop (WWC) diets. Animal categories: lactating cows (LC), dry cows (DC), heifers (HF), young animals (YA); § *Avena sativa* L. with *Vicia sativa* L.; # Vicia faba minor; ‡ *Pisum sativum* L.; * kg/head/day.

| Category | WWC1 | | | | WWC2 | | | | WWC3 | | | |
| --- | --- | --- | --- | --- | --- | --- | --- | --- | --- | --- | --- | --- |
| | LC | DC | HF | YA | LC | DC | HF | YA | LC | DC | HF | YA |
| **Feed kg/head/day** | | | | | | | | | | | | |
| Oat hay | - | - | - | - | 3.0 | - | 1.5 | 2.0 | 3.5 | 8.0 | 10.0 | 3.5 |
| Mixed meadow hay § | 10.0 | - | 5.0 | 2.8 | - | - | - | - | - | - | - | - |
| Straw | - | 7.5 | - | - | 1.5 | 8.5 | 2.0 | 1.0 | 2.0 | 3.0 | - | - |
| Maize silage | - | - | - | - | 15.0 | - | 3.0 | 1.5 | 15.0 | - | - | - |
| **Raw concentrate kg/head/day** | | | | | | | | | | | | |
| Maize grain/flour | - | - | - | - | 6.5 | 2.0 | 4.0 | 1.0 | 4.0 | - | - | - |
| Barley | - | - | - | - | - | - | - | - | - | - | - | - |
| Soybean meal | - | - | - | - | 2.9 | 0.8 | 2.0 | 0.5 | 1.0 | - | - | - |
| Fava bean # | - | - | - | - | - | - | - | - | 2.0 | - | - | - |
| Pea ‡ | - | - | - | - | - | - | - | - | 1.0 | - | - | - |
| **Market concentrate g/head/day** | | | | | | | | | | | | |
| Maize flour | 3200 | 750 | 1100 | 660 | - | - | - | - | - | - | - | 400 |
| Soybean seeds (dehulled/flaked) | 1800 | 400 | 400 | 240 | - | - | - | - | - | - | - | 100 |
| Soybean seeds (roasted) | - | - | 1100 | 660 | - | - | - | - | - | - | - | 80 |
| Barley | 1400 | - | - | - | - | - | - | - | - | - | - | - |
| Molasses | 400 | 50 | 120 | 70 | - | - | - | - | - | - | - | 50 |
| Bran | 200 | 900 | 360 | 220 | - | - | - | - | - | - | - | 270 |
| Palm oil | 100 | - | - | - | - | - | - | - | - | - | - | - |
| Sunflower meal | - | 300 | - | - | - | - | - | - | - | - | - | 250 |
| Beet pulp | - | 100 | 900 | 550 | - | - | - | - | - | - | - | - |
| Wheat flour shorts | - | - | - | - | - | - | - | - | - | - | - | 380 |
| **Chemical composition (%)** | | | | | | | | | | | | |
| Dry matter (DM) * | 16.0 | 9.5 | 7.9 | 4.6 | 17.5 | 10.3 | 9.6 | 4.5 | 17.1 | 9.8 | 8.9 | 4.5 |
| Crude protein (%DM) | 13.50 | 8.20 | 12.15 | 12.40 | 14.10 | 8.30 | 14.55 | 12.60 | 13.30 | 9.90 | 12.00 | 13.80 |
| Ether extract (%DM) | 3.60 | 2.15 | 3.20 | 3.30 | 5.30 | 3.40 | 5.75 | 5.55 | 4.30 | 2.75 | 3.00 | 3.45 |
| Crude fiber (%DM) | 22.25 | 32.60 | 22.20 | 21.80 | 22.30 | 33.50 | 21.60 | 23.00 | 25.35 | 36.70 | 35.00 | 29.65 |
| Ash (%DM) | 4.85 | 5.80 | 4.25 | 4.25 | 5.20 | 4.25 | 4.90 | 5.00 | 5.15 | 3.30 | 3.00 | 4.10 |

*2.3. Emissions*

Total emissions were estimated for the farms, according to the main difference that is wheat production, i.e., NWC and WWC. The fuel combustion, electricity consumption, enteric emissions, crop residue emissions, manure management and the emissions due to chemical fertilization were considered.

2.3.1. Enteric Emissions

The Refinement (2019) of the Intergovernmental Panel on Climate Change (IPCC) [16,17] methodology was adopted, considering the relationship between gross energy intake and emissions. According to the updated IPCC guidelines for the specific factors within the Tier 2 method, Equation (2) was adopted.

$$EF = [GE \times (Ym/100) \times x/55.56] \qquad (2)$$

where:

EF = emission factor, kg, $CH_4$ head$^{-1}$ x$^{-1}$;

GE = gross energy intake, MJ head$^{-1}$, provided by different ingredients, expressed as dry matter;

Ym = methane conversion factor, percent of GE in feed converted to methane. Ym in dairy buffaloes is assumed to be 6.5, as indicated in Table 10.12 (updated);

$x$ = the mean period, expressed in days, assumed for each livestock category, 90, 270 and 365 days, respectively, for dry cows, lactating cows and heifers. The enteric emissions of calves are not considered;

55.65 = the energy content of methane (MJ/kg $CH_4$).

The data on the GE supplied by different feed sources (e.g., hay, straw and soybean meal) were provided by INRAE [18]. The different percentages of feed were also considered for the assessment of the GE.

### 2.3.2. Methane Emissions from Manure Management

As suggested by the Refinement (2019) of the IPCC methodology, the Tier 2 method relies on two primary types of inputs that affect the calculation of methane emission factors from manure: the manure characteristics and the animal waste management system characteristics (AWMS). Equation (3) was also adopted to assess these emissions:

$$EF = (VS_T * x) \times [ B_{0(T)} \times 0.67 * \Sigma_{S,k} * MCF_{S,k} / 100 \times AWMS_{(T,S,k)}] \tag{3}$$

where:

EF = $CH_4$ emission factor for livestock category T, kg $CH_4$ animal$^{-1}$ days$^{-1}$;

$VS_T$ = daily volatile solid excreted for livestock category T, kg dry matter animal$^{-1}$ days$^{-1}$;

$x$ = basis for calculating annual VS production, days year$^{-1}$;

$B_{0(T)}$ = maximum methane producing capacity for manure produced by livestock category T, m$^3$ $CH_4$ kg$^{-1}$ of VS excreted. IPCC guidelines suggest a default value of 0.10 for dairy buffaloes (Table 10.16 (updated));

0.67 = conversion factor of m$^3$ $CH_4$ to kilograms $CH_4$;

$MCF_{(S,k)}$ = methane conversion factors for each manure management system S by climate region k, %. IPCC guidelines suggest a default value of 47% for animals kept in a paddock without distinguishing by climatic areas and relative humidity (Table 10.17 (updated)).

$AWMS_{(T,S,k)}$ = fraction of livestock category T's manure handled using manure management system S in climate region k, dimensionless. IPCC guidelines suggest default values of 3, 34 and 63%, respectively, for paddock, slurry and solid management in Western Europe (Table 10A.6 (updated)).

The VS content of manure and urine was estimated, in accordance with the Tier 2 method, with Equation (4) (updated):

$$VS = [GE \times (1 - DE\%/100) + (UE \times GE)] \times (1 - ASH/18.45) \tag{4}$$

where:

VS = volatile solid excretion per day on a dry-organic matter basis, kg day$^{-1}$;

GE = gross energy intake, MJ day$^{-1}$;

DE% = digestibility rate of the feed. Different amounts of feed were also considered for the assessment of the DE, also considering the livestock categories. Different feeding periods were considered: from birth to the 90th day (weaning ration) and up to a year for the calves; 365, 95 and 270 days for the heifers, dry and lactating cows. Similarly for the GE, the data provided by INRAE [18] were adopted for the digestibility;

(UE × GE) = urinary energy expressed as fraction of GE. Typically, 0.04 GE can be considered urinary energy excretion by most ruminants, and this value was adopted in the current study;

ASH = the ash content of manure calculated as a fraction of the dry matter feed intake, specifying the different sources, as indicated in Tables 3 and 4;

18.45 = conversion factor for dietary GE per kg of dry matter (MJ kg$^{-1}$).

### 2.3.3. $N_2O$ Emissions from Manure Management

Although the IPCC guidelines indicate some criteria useful to assess nitrogen excretion, we followed the approach reported by Romano et al. [7] because some studies showed that buffaloes have a greater efficiency of N utilization compared with cattle [19,20]. Consequently, we adopted the equations suggested by Patra et al. [21] aimed to estimate the nitrogen excretion (urinary and fecal) in buffaloes. The N intake was counted by knowing the crude protein amount distinguished by livestock categories and farms.

The $N_2O$ emissions were then assessed adopting the IPCC 2019 guidelines [16,17].

#### Direct $N_2O$ Emissions

As suggested by IPCC 2019 [15], the Tier 1A method was applied, adopting Equation (5) (updated).

$$N_2O_D \ (mm) = [\Sigma_S[\Sigma_{T,P}((N_{T,P} \times Nex_{T,P}) \times AWMS_{T,S,P}) + Ncdg(s)] \times EF_{3S}] \times 44/28 \quad (5)$$

where:

$N_2O_D$ (mm) = direct $N_2O$ emissions from manure management in the country, kg $N_2O$ year$^{-1}$, assuming for each livestock category 95, 270, and 365 days, respectively, for dry and lactating cows, heifers and female calves;

$N_{(T,P)}$ = number of heads in each livestock category T on the farm, for production system P;

$Nex_{(T,P)}$ = annual average N excretion per head of livestock category T, on the farm, for production system P, previously assumed with the equation before;

$AWMS_{(T,S,P)}$ = fraction of total annual nitrogen excretion for each livestock category T that is managed in the manure management system S in the country, also considering the production system P (Tier 1A approach), dimensionless. These default values are provided by Table 10A.6 (updated);

$Ncdg(s)$ = annual nitrogen input via co-digestate in the country, kg N yr$^{-1}$, where the system (s) refers exclusively to anaerobic digestion. Not included in this study;

$EF_{3(S)}$ = emission factor for direct $N_2O$ emissions from the manure management system S in the country, kg $N_2O$-N/kg N in the manure management system S; i.e., 0.01 and 0.002 for solid storage and paddock. These values are provided by Table 10.21 (updated) for solid storage [16] and by Table 11.1 (updated) for paddock [17]. In particular, for this manure management a value suggested for dry climates was adopted;

S = manure management system;

T = category of livestock;

44/28 = conversion of ($N_2O$-N) mm emissions to $N_2O$ mm emissions.

#### Indirect $N_2O$ Emissions

Specific equations are useful to estimate nitrogen losses; subsequently, these data will be used to assess the indirect emissions due to volatilization and leaching. The Tier 1A method is applied to estimate both losses and the following equations (Equations (6) and (7) (updated)) are adopted to assess N due to leaching and volatilization, respectively:

$$N_{\text{VOLATILIZATION-MMS}} = [\Sigma_S[\Sigma_{T,P}((N_{T,P} \times Nex_{T,P}) \times AWMS_{T,S,P}) + Ncdg(s) \times Frac_{gasMS(T,S)}]] \quad (6)$$

$$N_{\text{LEACHING-MMS}} = [\Sigma_S[\Sigma_{T,P}((N_{T,P} \times Nex_{T,P}) \times AWMS_{T,S,P}) + Ncdg(s) \times Frac_{leachMS(T,S)}]] \quad (7)$$

where:

$N_{\text{VOLATILIZATION-MMS}}$ = amount of manure nitrogen that is lost due to the volatilization of $NH_3$ and NOx, assuming for each livestock category 95, 270 and 365 days, respectively, for dry and lactating cows, heifers and female calves; kg N year$^{-1}$;

$N_{\text{LEACHING-MMS}}$ = amount of manure nitrogen that is lost due to leaching, assuming for each livestock category 95, 270 and 365 days, respectively, for dry and lactating cows, heifers and female calves; kg N year$^{-1}$;

$N_{(T,P)}$ = number of heads in each livestock category T on the farm, for production system P;

$Nex_{(T,P)}$ = annual average N excretion per head of livestock category T, on the farm, for production system P;

$AWMS_{(T,S,P)}$ = fraction of total annual nitrogen excretion for each livestock category T that is managed in the manure management system S in the country, also considering the production system P (Tier 1A approach), dimensionless;

$Ncdg(s)$ = annual nitrogen input via co-digestate in the country, kg N yr$^{-1}$, where the system (s) refers exclusively to anaerobic digestion. Not included in this study;

P = productivity class, high or low, of the system (Tier 1A approach);

$Frac_{gasMS(T,S)}$ = fraction of managed manure nitrogen for livestock category T that volatilizes as $NH_3$ and NOx in the manure management system S. The default value 0.12 is provided by Table 10.22 (updated) and is adopted for solid storage for "other animals" [16];

$Frac_{leachMS(T,S)}$ = fraction of managed manure nitrogen for livestock category T that is leached from the manure management system S. The default value 0.02 is provided by Table 10.22 (updated) and is adopted for solid storage for "other animals" [16].

The emissions of $N_2O$ due to the volatilization and leaching of manure were assessed with the following equations (Equations (8) and (9)), adopting the previously calculated N losses:

$$N_2O_G \text{ (mm)} = (N_{\text{VOLATILIZATION-MMS}} \times EF_4) \times 44/28 \tag{8}$$

$$N_2O_L \text{ (mm)} = (N_{\text{LEACHING-MMS}} \times EF_5) \times 44/28 \tag{9}$$

where:

$N_2O_G$ (mm) = indirect $N_2O$ emissions due to volatilization of N from manure management in the farm, assuming for each livestock category 95, 270 and 365 days, respectively, for dry and lactating cows, heifers and female calves, kg $N_2O$ year$^{-1}$;

$EF_4$ = emission factor for $N_2O$ emissions from atmospheric deposition of nitrogen on soils and water surfaces, kg $N_2O$-N (kg $NH_3$-N + NOx-N volatilized)$^{-1}$. The default value is 0.005 (dry climate) kg $N_2O$-N (kg $NH_3$-N + NOx-N volatilized)$^{-1}$ and is given in Chapter 11, Table 11.3 (updated) [17].

$N_2O_L$ (mm) = indirect emissions due to leaching and runoff from manure management in the farm, assuming for each livestock category 95, 270 and 365 days, respectively, for dry and lactating cows, heifers and female calves, kg $N_2O$ year$^{-1}$;

$EF_5$ = emission factor for $N_2O$ emissions from nitrogen leaching and runoff, kg $N_2O$-N/kg N leached and runoff. The default value of 0.011 kg $N_2O$-N (kg N leaching/runoff)$^{-1}$ is given in Chapter 11, Table 11.3 (updated) [16].

### 2.3.4. Ammonia

The ammonia emissions provided by the livestock were also calculated. In agreement with other studies [7,22], the emission factor 17/14 was adopted to estimate the $NH_3$ amount pertinent to each farm.

### 2.3.5. $CO_2$ Emissions from Livestock, Emissions from Crop, Soil Residues and Synthetic Fertilizers

The $CO_2$ emissions from livestock were not estimated because the annual net $CO_2$ emissions are assumed to be zero as the $CO_2$ photosynthesized by plants is returned to the atmosphere as respired $CO_2$ [16]. $N_2O$ and $CO_2$ emissions from soils and crop residues are included in each crop input selected in the SimaPro database, although several categories were modified and accordingly loaded. The $N_2O$ emissions provided by urea and ammonium nitrate were also considered with Tier 2 and Tier 1 methods, respectively, for direct and indirect emissions, distinguished by the following equations:

$$N_2O_{\text{DIRECT}}\text{-N} = (F_{SN} \times EF_{1i}) \tag{10}$$

where:

$N_2O_{DIRECT}$-N = direct $N_2O$–N emissions from N inputs provided by urea and ammonium nitrate applied to the soils, kg $N_2O$–N year$^{-1}$;

$F_{SN}$ = annual amount of synthetic fertilizer N applied to soils, kg N year$^{-1}$;

$EF_{1i}$ = emission factor for $N_2O$ emissions from N inputs, kg $N_2O$–N (kg N input)$^{-1}$. The default value for dry climates is available in Table 11.1 (updated) [17].

$$N_2O_{ATD}\text{-N} = (F_{SN} \times Frac_{GASF}) \times EF_4 \tag{11}$$

where:

$N_2O_{ATD}$-N = indirect amount of $N_2O$–N produced from atmospheric deposition of N volatilized from managed soils, kg $N_2O$–N year$^{-1}$;

$F_{SN}$ = annual amount of synthetic fertilizer N applied to soils, kg N year$^{-1}$;

$Frac_{GASF}$ = fraction of synthetic fertilizer N that volatilizes as $NH_3$ and NOx, kg N volatilized (kg of N applied)$^{-1}$. In Table 11.3 (updated), the default values for urea and ammonium nitrate-based emissions are available (0.15 and 0.05, respectively) [17];

$EF_4$ = emission factor for $N_2O$ emissions from atmospheric deposition of N on soils and water surfaces, [kg N–$N_2O$ (kg $NH_3$–N + NOx–N volatilized)$^{-1}$], available in Table 11.3 (updated) for dry climate, i.e., 0.005 [16].

$$N_2O_{LEACH\text{-}N} = (F_{SN} \times Frac_{LEACH}) \times EF_5 \tag{12}$$

$N_2O_{LEACH\text{-}N}$ = indirect amount of $N_2O$–N produced from leaching and runoff of N additions to managed soils where leaching/runoff occurs, kg $N_2O$–N year$^{-1}$;

$F_{SN}$ = annual amount of synthetic fertilizer N applied to soils, kg N year$^{-1}$;

$Frac_{LEACH\text{-}N}$ = fraction of all N added to managed soils in regions where leaching/runoff occurs that is lost through leaching and runoff, kg N (kg of N additions)$^{-1}$. In Table 11.3 (updated), the default value is 0.24 [17];

$EF_5$ = emission factor for $N_2O$ emissions from N leaching and runoff, kg $N_2O$–N (kg N leached and runoff)$^{-1}$. The default value is 0.11 in Table 11.3 (updated) [17].

The conversion of $N_2O$–N emissions to $N_2O$ emissions was obtained by the following equation:

$$N_2O = N_2O\text{-}N \times 44/28 \tag{13}$$

2.3.6. Emissions from Electricity and Diesel Fuel

The $CO_2$ equivalents, provided by the combustion of fossil fuels and from electricity use, were estimated considering the amount of diesel fuel and the kWh of electricity consumed for farm operations. The consumption and emission factor of natural gas adopted in the farms was excluded. The amount of purchased diesel was quantified through interviews, such as the electricity consumption. As suggested by ENAMA [23], a standard value of 0.85 kg per liter as diesel density and a 3.13 eq. emission factor to estimate $CO_2$ release from the combustion of 1 kg of diesel were adopted. As for the electricity mix, we used the Italian emission factor (0.47 eq.) adopted in other studies [1,4,7].

*2.4. Impact Assessment and Software*

The software SimaPro 8.01 PhD, Pré Consultants 2015 was employed to estimate the environmental impacts. Two methods were adopted to assess the impact categories: (i) EPD 2013 for GWP, computed according to the $CO_2$ equivalent factors in a 100-year time horizon, Acidification Potential (AP, g $SO_2$ eq) and Eutrophication Potential (EP, g $PO_4^{3-}$eq), and (ii) ReCiPe Midpoint (H) for Agricultural Land Occupation (ALO, m$^2$y) and Water Depletion (WD, m$^3$) [13,24,25]. Table 5 shows the impact categories and the characterization factors used to evaluate the potential environmental burden assessed with the EPD 2013 method.

**Table 5.** Characterization factors of the main elementary flows of the impact categories investigated with EPD 2013.

| Category | Main Elementary Flow | Characterization Factor | Source |
|---|---|---|---|
| Global Warming Potential, kg $CO_2$ eq | $CO_2$<br>$CH_4$<br>$N_2O$ | 1<br>28<br>265 | IPCC 2019 [15] |
| Acidification, kg $SO_2$ eq | $NH_3$<br>NOx<br>$SO_2$ | 1.6<br>0.76<br>1.2 | Huijbregts 1999 [25] |
| Eutrophication, kg $PO_4{}^3$ eq | $NO_3$<br>$P_2O_3$ | 0.1<br>3.06 | Heijungs et al. 1992 [26] |

As indicated by Gerssen-Gondelach et al. [27], Agricultural Land Occupation (ALO) is defined as the area of land needed to produce the FU chosen—in our research, 1 kg FPCM provided by buffalo. When the land occupation is not directly available with the inventory analysis of a study, these authors suggested the following equation:

$$\text{m}^2/\text{kg FPCM} = \text{total grassland or cropland requirement (Ha)} \times 10{,}000 \text{ (m}^2)/\text{kg FPCM} \qquad (14)$$

SimaPro Pré Consultants, thanks to the ReCiPe method, allows us to investigate the Water Depletion (WD), processing the raw materials based on studies about water scarcity (WS), water stress index (WSI) and water productivity. The main algorithms adopted in the software, developed by Hoekstra [28,29], considered the environmental relevance of water productivity and the water footprint in water-rich areas.

## 3. Statistical Analysis

We calculated the GWP, AC, EU, ALO and WD of each farm using the LCA model. Then, these data were analyzed by one-way (general linear model procedure) ANOVA, using the production system as an independent variable. The data were analyzed using the "stats" package of R software [30], and then the Tukey test was adopted, developed with the LTukey function of the "laercio" R package [31]. In agreement with Silva and Azevedo [32], the Tukey test was chosen for carrying out the comparison test as it is characterized by greater rigor than other post hoc statistical comparison tests and has a greater control of type I error.

## 4. Results and Discussion

Buffalo dairy farms also produce meat and crop commodities, thus the environmental impact should be shared among the co-products [33]. According to other studies focused on bovine milk and meat [7,24,34], the allocation criterion was adopted. In particular, the economic allocation resulted in a more suitable strategy applicable for buffalo farms. In agreement with Pirlo et al. [11], in this study, the economic allocation criterion was applied, which was preferred to mass allocation because it better represents the societal cause of buffalo farm environmental impacts.

### 4.1. Role of the Allocation

The economic allocation mitigated the impacts, mainly in the WWC system. This criterion was applied to determine the weight of the co-products, avoiding the expansion of system boundaries. The inputs related to the crop (seeds, land, fuels and agrochemicals) were loaded into the software without lightening them when considering straw, which is always reused (in WWC1, WWC2 and WWC3) as feed or litter. In the three WWC farms, all the grain is sold as food for human consumption and all the straw is reused, and the overproduced straw is stored as farm stock. As previously indicated, the meat provided by male calves was excluded, whereas culled cows were involved in both systems. As reported

by Mahath et al. [35], the following equation was adopted to subtract the equivalents from the functional unit (FU):

$$AF_{eco} = (V_{eco} \times P)_{milk} / \Sigma \, (V_{eco} \times P)_{milk, \, wheat \, grain, \, live \, weight \, culled \, cows} \tag{15}$$

where:

AF is the allocation factor;

Veco is the economic value (EUR/kg);

P is the total production on-farm (kg/year), milk as FPCM, wheat as harvested grain without straw; live-weight cattle as culled cows only.

The economic values of the milk, wheat grain and culled cows were obtained from interviews with the owners and were applied to evaluate the economic allocation. Economic values, mass amounts and percentages are reported in Table 6.

**Table 6.** Yields and economic incomes (%) derived from milk, wheat grain and culled cows (No Wheat Crop, NWC; With Wheat Crop, WWC).

|  | NWC1 | NWC2 | NWC3 | NWC Mean | WWC1 | WWC2 | WWC3 | WWC Mean |
|---|---|---|---|---|---|---|---|---|
| Milk income EUR/kg | 1.60 | 1.60 | 1.55 | 1.58 | 1.50 | 1.50 | 1.60 | 1.53 |
| Milk yield/year kg | 254,405 | 370,400 | 481,800 | 368,868 | 146,000 | 438,000 | 470,850 | 351,617 |
| Total milk income EUR | 407,050 | 592,640 | 746790 | 582,160 | 219,000 | 657,000 | 752930 | 542,977 |
| Wheat income EUR/kg | - | - | - | - | 0.33 | 0.30 | 0.27 | 0.30 |
| Wheat yield/year kg | - | - | - | - | 75,000 | 450,000 | 332,000 | 285,670 |
| Total wheat income EUR | - | - | - | - | 24,750 | 135,000 | 89,775 | 83,175 |
| Culled cows income EUR/head | 300 | 300 | 300 | 300 | 300 | 300 | 300 | 300 |
| Culled cows head/year | 8 | 20 | 25 | 18 | 5 | 16 | 16 | 12 |
| Total culled cows income EUR | 2400 | 6000 | 7500 | 5300 | 1500 | 4800 | 4800 | 3700 |
| Milk income % | 99.41 | 99.00 | 99.01 | 99.14 | 89.30 | 82.45 | 88.85 | 86.86 |
| Wheat income % | 0.00 | 0.00 | 0.00 | 0.00 | 10.09 | 16.94 | 10.59 | 12.54 |
| Culled cows income % | 0.59 | 1.00 | 0.99 | 0.86 | 0.61 | 0.60 | 0.57 | 0.59 |

*4.2. LCA Categories. Results and Discussion*

Table 7 shows the cradle-to-farm-gate life cycle results, related to the two different systems, i.e., NWC and WWC. The WWC group showed a total self-sufficiency in hay and straw supply, and in particular, the absence of purchased straw is related to the wheat crop. This arable area has significantly affected the Agricultural Land Occupation category as the production of wheat crop needs adequate land requirements. Additionally, WWC systems could also allow a significant reduction in eutrophication. These impacts exert a relevant effect on farm management, as shown by the PCA biplot (Figure 3), where the three WWC farms are grouped close together in the second quadrant. Although Berlese et al. [1] attributed to eutrophication a larger contribution of purchased feed than crops, in this study, the higher $PO_4^{3-}$ equivalents shown by the NWC system seem to find an explanation in the wider use of maize silage. Indeed, the comparison between NWC2/3 and WWC2/3 highlights the administration of corn silage in the NWC farms to all the livestock categories (Tables 3 and 4).

**Table 7.** Descriptive characteristics and Tukey's test.

| SYSTEM | FARMS | GWP kg $CO_2$ eq | AC g $SO_2$ eq | EU g $PO_4^{3-}$ eq | ALO $m^2y^{-1}$ | WD $m^3$ |
|---|---|---|---|---|---|---|
| No Wheat Crop | NWC1 | 4.65 | 50.23 | 13.27 | 8.52 | 2.26 |
| | NWC2 | 5.14 | 39.96 | 14.42 | 10.64 | 1.55 |
| | NWC3 | 4.95 | 37.81 | 15.62 | 11.56 | 1.56 |
| | mean NWC | 4.91a | 42.66a | 14.43a | 10.24b | 1.79a |
| | standard deviation | 0.24 | 6.63 | 1.17 | 1.56 | 0.41 |
| With Wheat Crop | WWC1 | 5.30 | 38.07 | 11.73 | 13.70 | 1.84 |
| | WWC2 | 5.51 | 30.75 | 11.21 | 15.69 | 1.57 |
| | WWC3 | 4.75 | 39.24 | 12.17 | 15.04 | 1.65 |
| | mean WWC | 5.19a | 36.02a | 11.70b | 14.81a | 1.69a |
| | standard deviation | 0.39 | 4.60 | 0.48 | 1.01 | 0.14 |

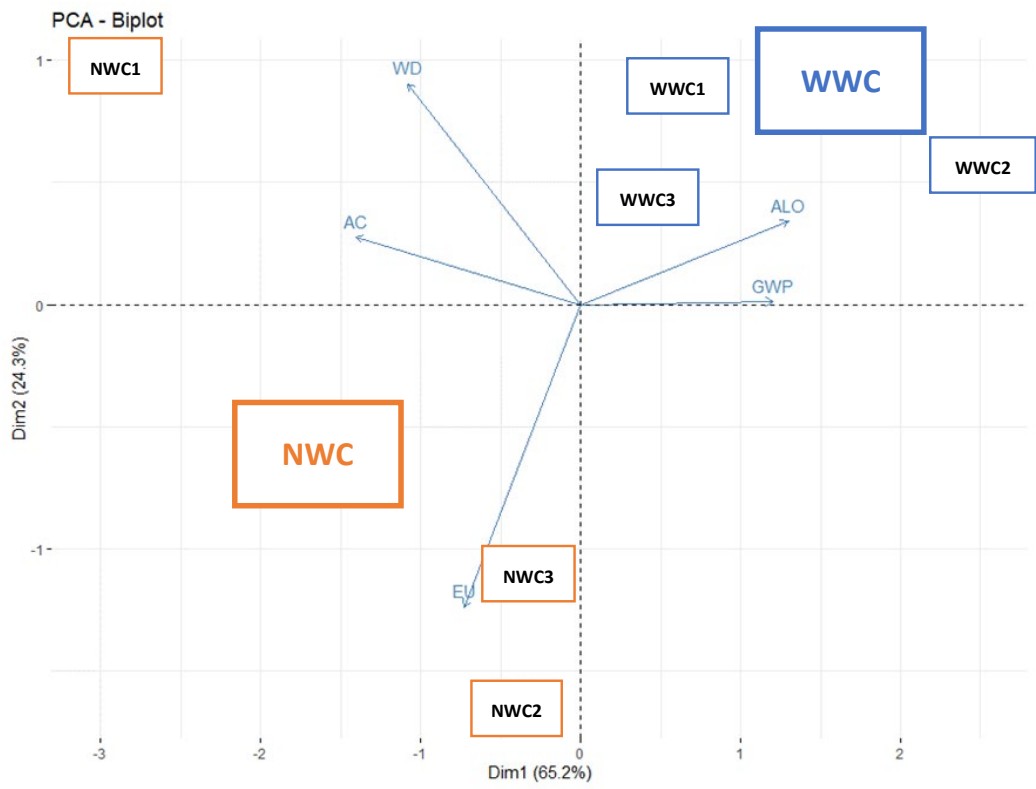

**Figure 3.** Biplot of the PCA analysis with the centroids of the two groups. Centroids are shown by bold boxes as WWC and NCW, respectively.

The multivariate analysis of the main components has shown how the choice of the variables that describe the present study, and which are represented by the LCA descriptors (GWP, AC, EU, ALO, WD), allows us to obtain two main components that explain altogether almost 90% of the variability of the overall dataset. In particular, the first component, set as the horizontal axis of the biplot, explains 65.2% of the overall variability, while the second component, set as the vertical axis of the biplot, explains 24.3% of the overall variability.

The biplot obtained from the PCA analysis (Figure 3) allows us to observe that the two variables that clearly discriminate the observed farms are the ALO and the GWP, placing the WWC farms on the right of the floor, well separated from the NWC farms on the left of the floor. The first farms are in fact characterized by higher ALO and GWP values than the second group.

Regarding the second component, we observed an influential effect of the EU variable between two NWC farms for higher values than WWC farms, creating a vertical separation.

Therefore, in the upper part of the biplot there are farms with lower EU values, while in the lower part of the biplot there are farms with higher EU values. In the upper left part, there is a farm belonging to the NWC group, which differs for higher values of WD and AC, but it must be remembered that the statistical analysis of variance did not show such statistically significant differences. The analysis then places the position of the centroids, defined as points whose coordinates are the average of the coordinates of the group elements obtained from the diametrically opposite components, top right for WWC farms and bottom left for NWC farms.

**GWP:** The impact on climate change was assessed using the IPCC characterization factors for the 100-year time horizon, according to the EDIP (2013) method, adopted in the study. As for climate stability, permanent grasslands store nearly as much carbon as forests (EIP-AGRI 2014) [36], and the carbon sequestration potential of permanent pastures was estimated between 0.01 and 0.3 Gt (gross tons of volume) C year$^{-1}$ [37,38], but all the farms involved in our research were without pasture. On other hand, the crop, field and fertilizer emissions were assessed with equations provided by IPCC guidelines [17].

Usually, the impacts are influenced by the weight of the functional unit (FU); for example, Berlese et al. [1] found a GWP of 6.4 kg $CO_2$ eq per kg of FPCM, higher than values reported by Pirlo et al. [11] (5.1 kg $CO_2$ eq per kg of FPCM) and also higher than our values (4.91 and 5.19 kg $CO_2$ eq for NWC and WWC, respectively). Indeed, the milk production observed by Berlese et al. [1] was 1409 kg FPCM/buffalo per lactation, in comparison to the mean value showed by Pirlo et al. [11] of 2251 kg FPCM/buffalo per lactation—a comparable amount with ours (2866 and 2000 kg FPCM/buffalo per lactation, respectively, for NWC and WWC systems). As suggested by Gerber et al. [39], the milk production of a farm is a crucial parameter because, in general, high productivity reduces environmental impact. Although NWC farms showed a higher milk yield than WWC farms, the $CO_2$ equivalents were not affected.

**AC:** In this study, no significant differences were found between the two systems. The values obtained in this study were very similar to those reported by Berlese et al. [1] ($37.3 \pm 3.97$ g $SO_2$ eq), and lower than those reported by Pirlo et al. [11] (65 g $SO_2$), where all the farms were corn silage-based.

A study focused on dairy buffaloes [4] compared a pasture-based system with a confined rearing system, showing that the main contribution for $SO_2$ equivalents was derived from the production of maize silage in the confined system. In agreement with these authors, our research showed that, despite the absence of differences between NWC and WWC farms for acidification potential, the higher trend for mean, minimum and maximum values in NWC farms could be explained by significant inputs related to the maize crop.

In the no-pasture-based system, the ammonium was identified by the authors as the first polluting substance followed by $SO_2$ and $NO_X$. Even though the research investigated bovine milk, Guerci et al. [40] found for grazing cattle (pasture only) very low $SO_2$ equivalents—7.44 g. This result may be attributed to the low use of inorganic fertilizers in pasture-based systems.

In addition, although the emissions of ammonia from manure management are recognized as significant contributors to acidification [4,41], the farms investigated in this study showed comparable manure handling, and the highest values observed in the NWC1 farm suggested a clarification. Research focused on dairy cattle [42] highlighted that the contributions to acidification were spilt among feed productions (both on- and off-farm), with a share of 31–57%. Moreover, Bragaglio et al. [12] found that the largest source of $SO_2$ equivalents is the production of concentrates. The purchasing of off-farm feed (total amount of concentrates, alfalfa hay, straw and more than 50% of hay meadow) concerning the NWC1 farm would explain the high AC value. In Figures 4 and 5, an explanatory comparison of feed sources between NWC1 and WWC1, the two farms without corn silage, is also given.

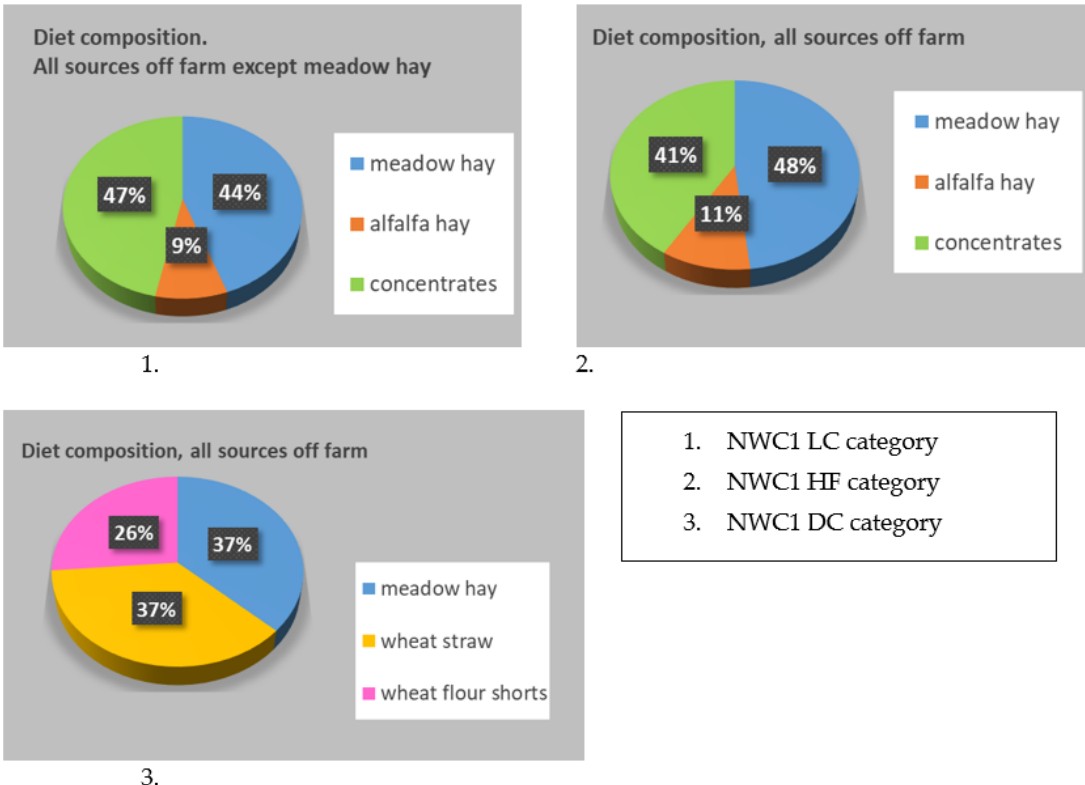

**Figure 4.** The graphs show the different percentage compositions of diets (lactating cows, LC; heifers, HF; dry cows, DC) in the No Wheat Crop 1 (NWC1) farm.

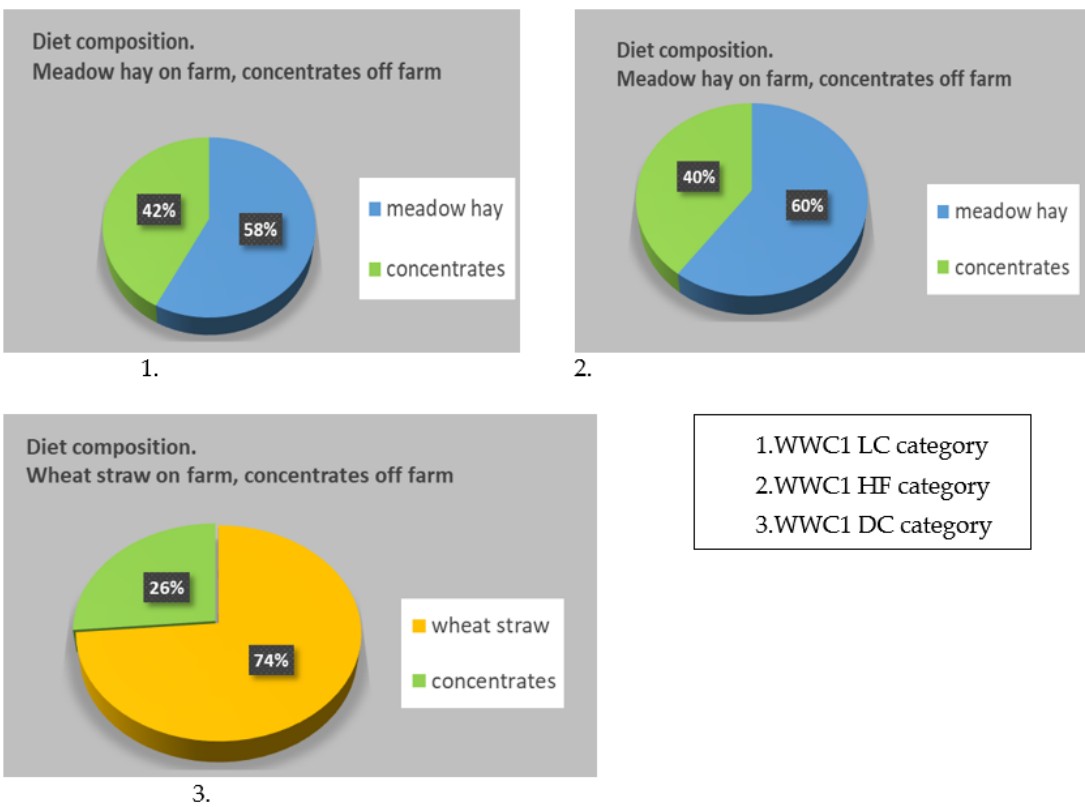

**Figure 5.** The graphs show the different percentage compositions of diets (lactating cows, LC; heifers, HF; dry cows, DC) in the With Wheat Crop 1 (WWC1) farm.

**EU:** This LCA descriptor, related to acidification, is affected by the conduction system and by environmental conditions, such as temperature, relative humidity or rainfall. Table 7 suggests that NWC farms showed significantly ($p < 0.05$) higher values (14.43 g $PO_4^{3-}$eq) than WWC farms (11.70 g $PO_4^{3-}$eq), which showed a more homogenous profile as a system (Figure 3). Sabia et al. [41] identified a functional unit (FU) of 1 kg of weight gain in the period needed to reach the age of puberty in buffalo heifers and compared animals kept in the pasture or in confinement. They reported higher values of eutrophication potential (g $PO_4^{3-}$) in the confined system. Similar findings, although focused on marine eutrophication (g N equivalents), were also reported by other authors [4]. The authors explained the different impacts because the main source of pollution was the production of corn silage, as also observed in dairy cattle by Bartl et al. [43].

As indicated in the third PCA biplot quadrant (Figure 3), the NWC2 and NWC3 farms, producing corn silage, are characterized by high $PO_4^{3-}$ equivalents. In comparison with NWC corn silage-based farms, WWC farms showed a different administration of this forage. In NWC2 and NWC3 farms, all the livestock categories are feed with corn silage, whereas in the WWC2 farm this is provided to lactating cows, heifers and young animals. Finally, in the WWC3 farm its administration is reserved only for lactating cows.

**ALO:** Table 7 shows that lower values were found in the NWC system ($p < 0.05$) compared to the WWC system. Many studies estimated the land use (LU), land use change (LUC) or the agricultural land occupation (ALO) impacts of organic vs. conventional or pasture-based vs. confined farming modes. Often, impact categories show higher values in organic or pasture-based systems, as the low yields (milk and forages) and management efficiency partially explain these results [7,43,44]. In this research, mean yields in NWC and WWC farms of 2866 and 2000 kg FPCM/buffalo per lactation, respectively, were recorded. In addition, WWC farms showed the lowest milk production, as shown in Table 2. The arable land of these farms is characterized by a high extent required for wheat crop (38, 66 and 35% for WWC1, WWC2 and WWC3, respectively); moreover, the fodder self-supply is also ensured by a significant hay area. Finally, the low number of lactating cows in the three WWC farms (Table 1), in addition to explaining the low productions, would suggest a less specialized management compared to the NWC mode. This aspect needs to be confirmed with studies involving a larger number of farms.

The adoption of economic allocation, useful to subtract the pollutants attributable to the co-produced wheat grain, has not allocated enough $m^2$ year$^{-1}$ equivalents, such as to identify the WWC system neither equally nor less impacting than the NWC one.

**WD:** This LCA category should be considered from a global perspective, since freshwater is a global resource, with growing global freshwater demand while global freshwater availability is limited [29]. In agreement with this framework, several studies are focused on the water footprint, distinguishing green water, blue water and grey water.

In our research, no significant differences were found comparing the NWC and the WWC system ($1.79 \pm 0.41$ vs. $1.69 \pm 0.14$ m$^3$). Sabia et al. [4] stated that no data are available on WD in dairy buffaloes, while conflicting results are available for dairy cattle. In Noord-Brabant (Netherlands), De Boer et al. [45] observed that 66 l of consumptive water was needed for 1 kg of FPCM, whereas at world level a water footprint of 1207 m$^3$/ton of milk [46] and a consumption of 544 l per 1 kg di FPCM [45] have been reported. In the other impact categories, dairy buffaloes [11] showed impacts 4-fold higher than dairy cattle [40] due to their lower milk production (roughly 4-fold lower even in terms of FPCM), even though input levels and categories and animal and farm dimensions were roughly similar.

In agreement with these studies [11,41], other authors found [47–50], in a similar Mediterranean environment, comparable results (0.38–0.55 m$^3$ and 0.52 m$^3$ per kg bovine FPCM, respectively), with a 4-fold ratio.

In the PCA biplot (Figure 3), the NWC1 farm showed, as indicated in Table 7, a different profile from the other NWC farms, with lower m$^3$ equivalents. Despite the absence of irrigated crops (corn), WD could be explained by information acquired from the interviews:

drinking water (human use) was indicated as an exclusive source, also adopted to clean the milking parlor. Although the NWC1 and WWC1 farms are characterized by a total absence of crop sources of grain (e.g., oat, barley, maize), additionally the NWC1 farm suffers from a lack of self-supply of forages (all the alfalfa and almost 50% of meadow hay were purchased).

## 5. Conclusions

This work performed an LCA of buffalo dairy farms, selected for similar general characteristics but different farming managements. A proportion of 1:2 of no maize silage and with maize silage was chosen as a system profile for both NWC and WWC farms. The main goal was the environmental impact assessment of dairy buffalo farming, according to forage self-sufficiency and combination with wheat crop production. WWC farms were characterized by a high availability of wheat straw, adopted as litter and administrated as fodder. In addition, a large arable area ensured a significant amount of hay; therefore, it was necessary to only purchase concentrates. The excess of straw and hay was managed with storage and, considering their low economic value, the allocation of the surplus would probably not have subtracted pollutants from the functional unit, i.e., FPCM. The productions of wheat crop entail adequate land requirements and the low milk yield observed in the WWC farms probably exacerbated the high values of Agricultural Land Occupation in agreement with other studies.

Interestingly, a high land occupation and low production, widespread in Mediterranean pasture-based systems, are often characterized by poor nutritional values of grazing grass.

The ALO results suggest that full self-sufficiency of forages would not be enough to describe a virtuous profile of the farms. Probably higher hay yields would have allowed the farms to obtain lower values of land occupation.

In order to estimate the possible reduction in the emissions due to the modification of some input variables, the performed analysis considers the possibility of reducing the fertilizers used in the agricultural stage. In particular, precision agriculture would allow targeted fertilization, leading to a reduction in the amount of agrochemicals for similar yields; a possible additional step could be to distinguish the different stages (tillage, fertilization, harvesting) with respect to the overall impact (midpoint categories). Afterwards, the base case could be compared with the improved scenario according to the normalized ReCiPe endpoint categories.

The $PO_4^3$ equivalents (EU) seemed to be mainly conditioned by the feeding strategies. Although in the life cycle assessment the impacts are often mitigated by high FU yields, this trend has not been highlighted in eutrophication.

As reported, dairy systems are often multifunctional processes; however, the allocation of by-products does not always mitigate the impacts. In particular, this study showed that a high availability of forages may have mitigated eutrophication (WWC farms), but it may have more heavily affected another disputed impact category: Agricultural Land Occupation.

**Author Contributions:** Conceptualization, P.D.P. and A.B.; methodology, A.B. and F.T.; software, A.B., E.R. and A.M.; validation, P.D.P. and E.R.; formal analysis, F.T. and E.R.; investigation, A.B.; resources, P.D.P. and A.M.; data curation, E.R.; writing—original draft preparation, E.R., P.D.P., F.T., A.M. and A.B.; writing—review and editing, F.T., E.R. and A.B.; visualization, P.D.P.; supervision, A.M. and F.T.; project administration, P.D.P. and A.B.; funding acquisition, A.B. All authors have read and agreed to the published version of the manuscript.

**Funding:** Andrea Bragaglio's research activity is granted by the European Union and Italian Ministry of Education, University and Research in the program PON 2014–2020 Research and Innovation, framework Attraction and International Mobility-1839894, Activity 1. The project was approved by the Italian Ministry of Education, University and Research.

**Institutional Review Board Statement:** Not applicable.

**Informed Consent Statement:** Not applicable.

**Data Availability Statement:**

a) CLAL, 2019. Production of Mozzarella di Bufala Campana PDO. Italy. https://www.clal.it/index.php/section=mozzarella_bufala_campana (accessed on 5 August 2021).

b) FAOSTAT, 2020. www.fao.org/faostat/en/#data (accessed on 5 August 2021).

c) INRAE, www.inrae.fr. https://mediatheque.inrae.fr/ (accessed on 5 August 2021).

d) ENAMA, 2005. Handbook of fuel consumption for agricultural subsidized use (Prontuario dei consumi di carburante per l'impiego agevolato in agricoltura). Available from: http://www.enama.it/php/pageflip.php_pdf_enama_int_prontuario.pdf&dir/it/pdf/monografie (accessed on 5 August 2021).

e) R Core Team. R: A Language and Environment for Statistical Computing; R Foundation for Statistical Computing: Wien, Österreich, 2019; Available online: https://www.R-project.org/ (accessed on 15 August 2021).

f) Laercio Junio da Silva. laercio: Duncan Test, Tukey Test and Scott-Knott Test. R Package Version 1.0-1. 2010. Available online: https://CRAN.R-project.org/package=laercio (accessed on 15 August 2021).

**Acknowledgments:** The authors are grateful to Giovanna Calzaretti and Francesco Giannico for their technical support. The authors are grateful to Laura Maresca, veterinary surgeon.

**Conflicts of Interest:** All authors declare that they do not have any conflicts of interest that could inappropriately influence this manuscript.

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
