# Peer review of "Dairy Buffalo Life Cycle Assessment (LCA) Affected by a Management Choice: The Production of Wheat Crop"

_sustainability, doi:10.3390/su131911108_

Round 1
Reviewer 1 Report
Detailed comments, recommendations and corrections are in the attached version of the manuscript.

Author Response
Dear reviewer,
in the attached file you can find the required changes.
Best regards

Reviewer 2 Report
Dear Authors,
your manuscript provides an interesting study on multifunctional processes of buffalo dairy sistem. In particular, you assessed the environmental impact of dairy buffalo farming, according to forages self-sufficiency and combination with wheat crop production. Manuscript is well written in all its chapters except for References. I suggest to read accurately references description in Instructions for Authors.
Please add:
* Cosentino, C.; Adduci, F.; Musto, M.; Paolino, R.; Freschi, P.; Pecora, G.; D'Adamo C.; Valentini, V. Low vs high “water footprint assessment” diet in milk production: A comparison between triticale and corn silage based diets. Emir. J. Food Agric. 2015, 27 (3), 312–317. doi: 10.9755/ejfa.v27i3.19226
In line 706 ……authors found [7,48, *], in a ….
** Freschi, P.; Musto, M.; Paolino, R.; Cosentino, C. Grazing and biodiversity conservation: Highlights on a natura 2000 network site. In The Sustainability of Agro-Food and Natural Resource Systems in the Mediterranean Basin, Vastola, A., Ed.; Springer International Publishing AG, Basel, Switzerland, 2015; pp. 271–288. doi: 10.1007/978-3-319-16357-4_18
In line 735 grazing grass [**]
Author Response

(The authors gave the same response as above.)

Reviewer 3 Report
In this paper, the authors performed an LCA analysis in Italian dairy buffalo farms. The paper is interesting and can be published after some improvements.
At some points, the journal template has been skipped. The first page is indicated as 5 of 19; the following ones as 4 of 23.
In line 42, “these authors” has to be substituted with the name of the first author et al.
The equation indicated in lines 89 and 90 should be equation (1).
In Figure 2, there is a black arrow that can be removed.
In line 548, the authors introduced a PCA biplot, which may not be familiar to all readers of the journal. A couple of lines explaining how points and vectors can be represented in this type of diagram might be helpful.
In lines 737 and 738, the authors suggested a way to reduce the land occupation. This sentence can be quickly supported, improving the quality of the article. The authors could do a scenario analysis to verify the effect of the hay yield on ALO. Similar analyses were performed in previous papers: doi: 10.1016/j.supflu.2017.11.005; doi: 10.1016/j.scitoenv.2018.12.131.
Author Response

(The authors gave the same response as above.)
